# Analysis of the Spread of COVID-19 in the USA with a Spatio-Temporal Multivariate Time Series Model

**DOI:** 10.3390/ijerph18020774

**Published:** 2021-01-18

**Authors:** Rongxiang Rui, Maozai Tian, Man-Lai Tang, George To-Sum Ho, Chun-Ho Wu

**Affiliations:** 1School of Statistics, Renmin University of China, Beijing 100872, China; raynerrui@ruc.edu.cn; 2College of Medical Engineering and Technology, Xinjiang Medical University, Ürümqi 830011, China; mztian@ruc.edu.cn; 3Department of Mathematics, Statistics and Insurance, Hang Seng University of Hong Kong, Hong Kong, China; 4Department of Supply Chain and Information Management, Hang Seng University of Hong Kong, Hong Kong, China; georgeho@hsu.edu.hk (G.T.-S.H.); jackwu@ieee.org (C.-H.W.)

**Keywords:** columnar density of total atmospheric ozone, COVID-19, maximum temperature, minimum temperature, spatio-temporal multivariate time-series analysis, USA

## Abstract

With the rapid spread of the pandemic due to the coronavirus disease 2019 (COVID-19), the virus has already led to considerable mortality and morbidity worldwide, as well as having a severe impact on economic development. In this article, we analyze the state-level correlation between COVID-19 risk and weather/climate factors in the USA. For this purpose, we consider a spatio-temporal multivariate time series model under a hierarchical framework, which is especially suitable for envisioning the virus transmission tendency across a geographic area over time. Briefly, our model decomposes the COVID-19 risk into: (i) an autoregressive component that describes the within-state COVID-19 risk effect; (ii) a spatiotemporal component that describes the across-state COVID-19 risk effect; (iii) an exogenous component that includes other factors (e.g., weather/climate) that could envision future epidemic development risk; and (iv) an endemic component that captures the function of time and other predictors mainly for individual states. Our results indicate that maximum temperature, minimum temperature, humidity, the percentage of cloud coverage, and the columnar density of total atmospheric ozone have a strong association with the COVID-19 pandemic in many states. In particular, the maximum temperature, minimum temperature, and the columnar density of total atmospheric ozone demonstrate statistically significant associations with the tendency of COVID-19 spreading in almost all states. Furthermore, our results from transmission tendency analysis suggest that the community-level transmission has been relatively mitigated in the USA, and the daily confirmed cases within a state are predominated by the earlier daily confirmed cases within that state compared to other factors, which implies that states such as Texas, California, and Florida with a large number of confirmed cases still need strategies like stay-at-home orders to prevent another outbreak.

## 1. Introduction

The pandemic due to the coronavirus disease 2019 (COVID-19), caused by the novel severe acute respiratory syndrome coronavirus 2 (SARS-CoV-2) [1], was the most disastrous incident in 2020, causing millions of deaths and resulting in economic activity worldwide falling sharply. According to the latest report of the World Health Organization (WHO), the cumulative cases around the world reached 28,637,952 and the cumulative deaths were 917,417 as of 13 September 2020 (https://www.who.int/docs/default-source/coronaviruse/situation-reports/20200914-weekly-epi-update-5.pdf?sfvrsn=cf929d04_2). Furthermore, the World Bank suggested that most countries would be expected to suffer from economic recession in 2020 (https://www.worldbank.org/en/news/feature/2020/06/08/the-global-economic-outlook-during-the-covid-19-pandemic-a-changed-world).

Although some countries and regions (e.g., North America, China, and Europe) are actively developing vaccines with some showing encouraging signs [2,3,4], it is almost impossible to provide sufficient effective vaccines to every person in the next few years. Hence, this pandemic will undeniably last several months or even a few years. As one of the most developed countries in the world, the United States declared a public health emergency on 31 January 2020, and preventive and proactive measures (e.g., suspending the entry and the quarantine of foreign nationals seeking entry) have been taken to control the spread of the virus and treat those affected. However, it has become one of the most severely affected nations as the respective numbers of confirmed cases and deaths approximately account for 1/5 (i.e., 6,386,832 and 191,809) of the whole global cases as of 13 September 2020. Even worse, as Chowell and Mizumoto [5] argued, states and territories with the largest proportions of older populations (such as Florida, Maine, and Puerto Rico) have become the places with the largest number of confirmed cases. The spread of this pandemic in the USA has become a global concern.

Typically, susceptible-infected-recovered (SIR) based models (e.g., SIRD [6]), first proposed by Kermack and McKendrick [7], are widely used models due to their simplicity and good performance. However, these models (such as susceptible-infected-recovered-susceptible (SIRS), susceptible-exposed-infected-recovered (SEIR), and susceptible-exposed-infected-recovered-susceptible (SEIRS)) only take into account the tendency of the related epidemic transmission corresponding to one single region, and other useful information can hardly be uncovered, which includes the impacts derived from the place itself, other areas, and other exogenous variables. Some other models that are utilized to characterize epidemic pervasion are based on time series models. For instance, seasonal autoregressive integrated moving average (SARIMA) models were employed for modeling infectious disease count data in Helfenstein [8] and Trottier et al. [9]. Recently, different time series models (e.g., auto-regressive integrated moving average (ARIMA), the Holt–Winters additive model, and HWAAS) and machine learning approaches (e.g., Prophet, DeepAR, and N-Beats) have been adopted to analyze and compare the prediction accuracy of the percentage of active cases per population based on the COVID-19 data from ten countries with the highest number of confirmed cases as of 4 May 2020 [10].

Held et al. [11] proposed a space-time multivariate time series model (denoted as the HHHmodel) that can be applied to model multiple-unit cases where the “unit” can be different geographical regions, different age groups, or different epidemics caused by different pathogens. Motivated by the HHH model, Paul et al. [12] and Paul and Held [13] developed a spatio-temporal framework to jointly model several epidemics by considering the spatial interaction effect, as well as the time autoregressive effect. Their models have been applied to analyze the transmission of dengue fever in Guandong Province in China in 2014 [14], malaria and cutaneous leishmaniasis analysis in Afghanistan [15], hemorrhagic fever with renal syndrome in Zhejiang Province of China [16], and the effect of containment measures for COVID-19 in Italy [17].

One drawback of the model proposed by Paul et al. [12] and Paul and Held [13] is that it mainly takes care of the connection between the current number of infected cases and the previous numbers of infected cases and the adjacent units/areas, which may ignore other exogenous predictors. As a result, it is limited in its interpretability and applicability for infectious diseases such as COVID-19. In fact, recent studies (e.g., [18,19]) have pointed out that some weather/climate related variables show statistically significant associations with the transmission of COVID-19.

To overcome this limitation for the analysis of state-level time series of the COVID-19 contagion effect, we consider a spatio-temporal framework based on the multivariate time series model proposed by Paul et al. [12] and Paul and Held [13], which however decomposes the COVID-19 risk additively into autoregressive, spatiotemporal, exogenous, and endemic components. Briefly, the autoregressive and spatiotemporal components respectively describe the within-state and across-state COVID-19 risk effects. The exogenous component includes other factors that could affect future epidemic development risk, while the endemic component captures the function of time and other predictors mainly about individual states.

Briefly, some weather/climate related variables are carefully selected as exogenous factors in our analyses. Indeed, some climate/weather related variables have been shown to be correlated with epidemic transmission in the relevant literature. For instance, in a study of the influence of weather on the foot-and-mouth disease epidemic spread from 1967 to 1968, Hugh-Jones and Wright [20] argued that wind and precipitation played a major role in the spread of the disease, especially wind. According to Tan et al. [21], the environmental temperature can influence the spread of SARS. Qi et al. [18] found that the daily average temperature and daily average relative humidity are significantly negatively associated with the daily confirmed cases of COVID-19 in Hubei, China. Similarly, Tosepu et al. [19] found that the average seasonal temperature was significantly correlated with COVID-19 in Jakarta, Indonesia.

The rest of this article is organized as follows. In Section 2, we elaborate our spatio-temporal multivariate time series model, including the sub-models of each coefficient in the model. In Section 3, we employ our model for analyzing the COVID-19 count data of the USA and show our main findings. In Section 4 and Section 5, the discussion and conclusions are presented, respectively. We report the technical materials for parameter estimation in Appendix A.

## 2. Development of the Model

### Models

Let $Y_{r,t}$ denote the number of infected cases in state *r* at time point *t* with r=1,⋯,R,t=1,⋯,T. Usually, $Y_{r,t}$ is assumed to follow a Poisson distribution [22,23,24]. Since the number of infected cases in each state is hardly totally observed (i.e., the existence of heterogeneity for different states), employing the Poisson assumption could underestimate the underlying dispersion. Here, we adopt the negative binomial distribution [11,12,14]. That is, suppose $Y_{r,t}$ follows a conditional negative binomial distribution, i.e., $Y_{r,t}|Y_{\cdot ,t-l},V∼\mathrm{NegBin}\left(\mu_{r,t},\varepsilon_{r}\right)$ for r=1,⋯,R,t=1,⋯,T, with conditional mean $\mu_{r,t}$ and conditional variance:$$\sigma_{r,t}^{2}=\mu_{r,t}\left(1+\varepsilon_{r}\mu_{r,t}\right),$$
where $Y_{\cdot ,t-l}$ indicates the vector consisting of the number of infected cases of all states at time point t−l, *l* is the time lag term satisfying l∈{1,⋯,T−1}, $\varepsilon_{r}$ is the overdispersion parameter of state *r*, and V is a random effect vector with V∼N(0,Σ) with $\Sigma =\mathrm{diag}\left\{\sigma_{\left(\lambda \right)}^{2},\sigma_{\left(\psi \right)}^{2},\sigma_{\left(\theta \right)}^{2},\sigma_{\left(\zeta \right)}^{2}\right\}⊗I_{R\times R}$, ⊗ being the Kronecker product and $I_{R\times R}$ an R×R identity matrix. It is easy to see that when $\varepsilon_{r}$ equals zero, the distribution of $Y_{r,t}$ reduces to a Poisson distribution, whereas the larger the value of $\varepsilon_{r}$, the greater the overdispersion is. Thus, comparing with the Poisson assumption, the negative binomial assumption has wider applicability.

To embed other predictors in the distribution of $Y_{r,t}$, a hierarchical modeling procedure is employed here. In the first layer, the conditional mean $\mu_{r,t}$ is formulated as follows:(1)$$\mu_{r,t}=\lambda_{r,t}Y_{r,t-l}+\psi_{r,t}\Psi_{r,t-l}+\theta_{r,t}\Theta_{r,t-l}+\zeta_{r,t},$$
where:$$\begin{array}{cc} \Psi_{r,t-l}= & \sum_{j\to r}\omega_{r,j}Y_{j,t-l},\;\Theta_{r,t-l}=\sum_{j}\eta_{r,j}x_{j,k,t-l},\;j=1,\cdots ,R,\;k=1,\cdots ,K. \end{array}$$

Here, $x_{j,k,t}$ denotes the observation at time *t* of the *k*-th exogenous factor in state *j*, which could have an influence on state *r*. $\eta_{r,j}$ is an indicator with the value being $1/m_{j}$ if $x_{j,k}$ has influence on state *r* and zero otherwise, where $m_{j}$ is the number of factors that have an influence on the number of cases of the *j*-th state. j→r indicates that states *j* and *r* are neighbors that share the same border. $\omega_{r,j}$ is an indicator with the value being $1/n_{j}$ if state *r* is adjacent to state *j* and zero otherwise, where $n_{j}$ is the number of states that have a common border with state *j*. Other choices of weights (i.e., $\omega_{r,j}$) are also available in [12,13,14] and the references therein.

According to Giuliani et al. [25], $\lambda_{r,t}Y_{r,t-l}$, $\psi_{r,t}\Psi_{r,t-l}$, and $\zeta_{r,t}$ are respectively called the epidemic-within component, epidemic-between component, and endemic component. In this paper, we adopt the terminology from Giuliani et al. [25], and we further call $\theta_{r,t}\Theta_{r,t-l}$ the epidemic-boosted component. It is noteworthy that $\Psi_{r,t-l}$, which is based on the space-time dimension, mainly includes the interaction information between one state and other states neighboring that state, while $\Theta_{r,t-l}$ contains the correlation information of other exogenous factors between one state and other states neighboring that state. Comparing with the model proposed in Paul et al. [12] and Paul and Held [13], our proposed model in (Equation 1) improves the interpretability, as well as the applicability.

In the second layer, for parameters $\lambda_{r,t},\psi_{r,t}$, and $\zeta_{r,t}$, we adopt the same strategy as given in Paul et al. [12] and Paul and Held [13]; that is, each parameter assumes the following log-linear form:(2)$$\log\left(\cdot_{r,t}\right)=\alpha_{r}^{\left(\cdot \right)}+V_{r}^{\left(\cdot \right)}+{\beta^{\left(\cdot \right)}}^{⊤}z_{r,t}^{\left(\cdot \right)},$$
where $V^{\left(\cdot \right)}$ is assumed to have a multivariate normal distribution with zero mean and covariance matrix $\sigma_{\left(\cdot \right)}^{2}I_{R\times R}$, i.e., $V^{\left(\cdot \right)}∼N\left(0,\sigma_{\left(\cdot \right)}^{2}I_{R\times R}\right)$. We further discuss the formulation of each parameter as follows.

We first consider autoregressive parameter $\lambda_{r,t}$. As suggested in Paul et al. [12], Cheng et al. [14], Adegboye et al. [15], and Wu et al. [16], $\lambda_{r,t}$ is formulated by the following log-linear form:(3)$$\log\left(\lambda_{r,t}\right)=\alpha^{\left(\lambda \right)}+V_{r}^{\left(\lambda \right)},$$
where $\alpha^{\left(\lambda \right)}$ is related to the intercept term and $V_{r}^{\left(\lambda \right)}∼N\left(0,\sigma_{\left(\lambda \right)}^{2}\right)$.

For $\psi_{r,t}$, it satisfies:(4)$$\log\left(\psi_{r,t}\right)=\alpha^{\left(\psi \right)}+V_{r}^{\left(\psi \right)}+\beta_{1}^{\left(\zeta \right)}\log\left(Pu_{r,t}\right),$$
where $\alpha^{\left(\lambda \right)}$ is related to the intercept term and $V_{r}^{\left(\lambda \right)}∼N\left(0,\sigma_{\left(\lambda \right)}^{2}\right)$. For the choice of $Pu_{r,t}$, unlike Paul et al. [12] and Paul and Held [13], Giuliani et al. [25] argued that it should be a variable that reflects the possible heterogeneous influence for different regions, and their choice was the population of a state. However, we believe that people of different ages could be significantly divergent under the consideration of the infection effect from the population. In this regard, we define $Pu_{r,t}$ as the population size of people whose ages are under 65 in state *r* at time *t* based on the fact that this group of people is more likely to travel to other places.

For $\theta_{r,t}$, the log-linear formula may not be suitable as the influence of such exogenous variables could have a positive or negative effect on epidemic transmission, i.e., the sign of $\theta_{r,t}$ could be “+” or “−”. Here, we suppose that $\theta_{r,t}$ follows a normal distribution with mean $\alpha^{\left(\theta \right)}$ and variance $\sigma_{\left(\theta \right)}^{2}$, i.e.,
(5)$$\theta_{r,t}=\alpha^{\left(\theta \right)}+V_{r}^{\left(\theta \right)},$$
where $V_{r}^{\left(\theta \right)}∼N\left(0,\sigma_{\left(\theta \right)}^{2}\right)$.

Giuliani et al. [25] employed a second-order polynomial log-linear regression to evaluate the fluctuation of the number of confirmed cases from the perspective based on the time dimension. This is reasonable when the epidemic is in the early stage (i.e., *t* is relatively small). However, with the development of the pandemic, the reproduction number will inevitably tend to be small as the population for a specific state is limited. Thus, we suggest to use the s-shaped growth curve—logistic growth model—which was also employed to study age-specific case-fatality rates of COVID-19 in China and Italy [26]. That is, we consider:(6)$$\begin{array}{cc} \log(\zeta_{r,t})= & \alpha^{\left(\zeta \right)}+V_{r}^{\left(\zeta \right)}+\beta_{1}^{\left(\zeta \right)}\log\left(\mathrm{logit}(t)\right)+\beta_{2}^{\left(\zeta \right)}\log\left(Po_{r,t}\right), \end{array}$$
where:$$\mathrm{logit}\left(t\right)={\left(1+\exp\left\{-(\beta_{3}^{\left(\zeta \right)}+\beta_{4}^{\left(\zeta \right)}t)\right\}\right)}^{-1}$$
is a logistic function and $Po_{r,t}$ is defined as the population size of people whose ages are over 65 years old in state *r* at time *t*.

## 3. Results

### 3.1. Data of Interest

Study area: Here, we consider the 50 states plus Washington, D.C. (DC), for our COVID-19 analyses. However, American Samoa, Guam, the Northern Mariana Islands, the Commonwealth of Puerto Rico, and the Virgin Islands are excluded from our study for simplicity.

COVID-19 data: We obtained the state-level confirmed cases data on COVID-19 in the USA from Kaggle, which are available from https://www.kaggle.com/sudalairajkumar/covid19-in-usa. Here, we are mainly interested in the cumulative positive cases, as this will be used for the calculation of the number of daily increased COVID-19 cases in the USA. On 14 March 2020, the U.S. President held a coronavirus conference, one day after he declared the pandemic a national emergency (https://www.rev.com/blog/transcripts/donald-trump-coronavirus-press-conference-transcript-march-14). For this reason, we consider those data starting from March 15th, 2020.

Weather/climate data: The state-level weather and climate data we use in this paper are openly available from Kaggle, which is fully powered by Dark Sky and can be downloaded from https://www.kaggle.com/eeemonts/weatherclimate-data-covid19?select=csv. The factors included in our analyses are the maximum temperature (MaT), minimum temperature (MiT), humidity (Hu), the probability of precipitation appearance (PA), the percentage of cloud coverage (CC), sea-level air pressure (AP), wind speed (WS), and the columnar density of total atmospheric ozone (CDTAO). Consistent with the COVID-19 data, we only consider weather/climate data starting from March 15th, 2020.

Population data: Both the state-level population data of 2019 and the state-level population percentage of people over 65 years old were collected from Population Reference Bureau (PRB), which is available from (https://www.prb.org/usdata/indicator/age65/snapshot). Since the direct accessibility of the population over 65 years old is denied, we simply used the state-level population of 2019 multiplying the related percentage over 65 years old to get the approximation of the state-level population over 65.

Figure 1 depicts the daily confirmed cases and the cumulative confirmed cases in the USA. One can see that the cumulative confirmed cases and daily confirmed cases in some states such as Connecticut (CT), New Jersey (NJ), and New York (NY) have eased up; some states like North Dakota (ND) became more and more severe; and others tended to recur.

Figure 2 shows state-level subplots of the cumulative confirmed cases on September 15th, the daily confirmed cases on September 15th, the total populations in 2019, the populations over 65, and the populations under 65. It appears that both the cumulative confirmed cases and the daily confirmed cases have a strong consistency with the populations under 65, as well as the populations over 65. The appearance of a high infection rate among individuals under 65 could mean that there is a trend of younger people having severe COVID-19 infections in the USA as warned by Kass et al. [27], which can cause much worse situations (e.g., infecting more older adults) [28].

### 3.2. Weather and Environmental Factors’ Selection

To explore the association between weather/climate based factors and COVID-19 transmission, Tosepu et al. [19] applied the Spearman-rank correlation test, while Bashir et al. [29] utilized the Kendall and Spearman-rank correlation tests. Here, we first use the Kendall and Spearman-rank correlation tests to identify factors that have a significant correlation with daily confirmed cases.

Figure 3 shows the results from the Spearman and Kendall tests. According to the scatter plots, the *p*-values associated with MaT, MiT, Hu, CC, and CDTAO with respect to all states are mostly below 0.10, whereas those associated with AP and PA are mostly over 0.10. On the other hand, results from bar plots suggest that no particular factor has a remarkable association with the majority of the states. Briefly, AP and PA only have an influence on less than 25 states, while MaT, MiT, and CDTAO have a strong association with more than 40 states. According to Figure 4, it can be safely concluded that MaT, MiT, and CDTAO are the major factors that contribute to the strong associations in most of the states and are taken into account for further modeling analyses.

### 3.3. Optimal Parameters’ Determination

For better fitting the confirmed cases in different states, the optimal time lag *l* needs to be determined. Considering that the desirable range for *l* is dynamically increased, point-wise optimization seems too tedious and less efficient. Thus, we adopted the possible range from one to 14, which is the time interval that the CDC suggested to stay at home after one’s last contact with a person who has COVID-19 (https://www.cdc.gov/coronavirus/2019-ncov/if-you-are-sick/quarantine.html). Besides, all weather/climate factors were respectively nondimensionalized by Studentization to mitigate the impact of the different units.

Table 1 summarizes the correlation between different time lags and the related penalized log-likelihood values. It is noticed that when l=2, the average estimate of l(π,v) has the largest value (i.e., 48,510,611.518) with the smallest sd (i.e., 301,043.787). Hence, l=2 will be used in all subsequent analyses.

Table 2 shows the estimates of π and σ based on the configuration for time lag l=2. From the estimates of the dispersion parameters ε’s, we observe that the daily confirmed cases of COVID-19 show obvious overdispersion in almost every state (especially in New Mexico (NM)), which confirms that using the negative binomial distribution is a sensible choice for analyzing the transmission of COVID-19 in the USA.

The estimates of σ characterize the heterogeneity of COVID-19 transmission across states. According to the results in Table 2, there is spatial variation concerning the epidemic-within component with ${\hat{\sigma }}_{\lambda }=0.851$, the epidemic-between component with ${\hat{\sigma }}_{\left(\psi \right)}$ = 0.836, the epidemic-boosted component with ${\hat{\sigma }}_{\left(\theta \right)}=0.816$, and the endemic component with ${\hat{\sigma }}_{\zeta }=0.853$. Therefore, we believe that there is significant spatial heterogeneity in the epidemic-within, endemic-between, epidemic-boosted, and endemic component.

### 3.4. Components Analysis

Figure 5 shows the state-level estimated random effects with respect to epidemic-within component, epidemic-between component, epidemic-boosted component, and endemic component. Clearly, heterogeneity appears in all components, and the random effects from the four components have a significant effect in most of the states. Here, we mainly focus on random effects from the epidemic-within component and epidemic-boosted component. From Subplot (a) in Figure 5, the estimates of $\alpha^{\left(\lambda \right)}+V^{\left(\lambda \right)}$ for most states are smaller than zero, which implies that community-level spread in most states has considerably alleviated from the perspective of epidemic-with component. Similarly, the estimates of $\alpha^{\left(\lambda \right)}+V^{\left(\lambda \right)}$ are smaller than the negative estimate of $\alpha^{\left(\theta \right)}$ (see Subplot (c) in Figure 5), which indicates higher values of weather/climate factors corresponding to less daily confirmed cases.

Figure 6 and Figure 7 show the state-level percentage of daily confirmed cases of each component in every single day and the state-level means of fitted values based on (Equation 1). It is obvious that the estimated values of the epidemic-within component are the highest among all components in most of the states (e.g., CA) as time goes on. This phenomenon suggests that cross-state spread, weather/climate factors’ influence, or other unobserved factors have little impact on COVID-19 transmission, whereas the previous state’s related cases predominate the fluctuation of future infections in most states. For several states such as Connecticut (CT), New Hampshire (NH), and Rhode Island (RI), no particular component demonstrates a dominant effect on daily confirmed cases. According to Figure 1, we find that with relatively less confirmed cases, the endemic component seems to be more dominant in these states, which is also shown in CT, NH, and RI.

## 4. Discussion

In this article, we find that MaT, MiT, and CDTAO have statistically significant associations with daily confirmed cases in almost all the states in America, based on both the Kendall and Spearman-rank correlation tests. Furthermore, from the estimated coefficients of the epidemic-boosted component, we identify that this association is negative. That is, higher MaT, MiT, and CDTAO correspond to smaller daily confirmed cases. However, further analysis uncovers that the previous daily confirmed cases in one state itself are generally predominant for the next confirmed cases, which suggests that states with a large number of confirmed cases tend to cause more infections.

Recent research [18,19] has shown some evidence of the correlation between weather/climate factors and COVID-19 transmission. Our work, based on an extended multivariate time series model, further confirms and quantifies the existence of a similar relationship. Unlike some existing models (e.g., [10]), our model successfully facilitates the interpretability and practicability by an additional term that characterizes the degree of the influence from some other external factors. However, one obvious drawback of our model is that a large number of unknown parameters need to be estimated, and the computational cost is therefore high. As a result, the effective sample size also needs to be sufficiently large. Recently, Kimball et al. [30] investigated a COVID-19 outbreak in a long-term care skilled nursing facility (SNF) in King County, Washington, recognized on 28 February 2020, and discovered that screening the SNF residents based on the symptoms related to this epidemic could not to discover all SARS-CoV-2 infections since they found that 23 (30.3%) workers had SARS-CoV-2 positive tests even if they were asymptomatic or presymptomatic on the testing day. Such clustering based infections dramatically aggregate the transmission intensity, making the analysis based on not only our model, but others previously mentioned non-trivial challenges. More recently, Rader et al.’s analysis [31] unveiled that “epidemics in crowded cities are more spread over time, and crowded cities have larger total attack rates than less populated cities”. Such phenomena further imply that an area with a higher population density could cause a much severer outbreak, which is also a situation that requires further investigation.

## 5. Conclusions

As COVID-19 has become the most disastrous health event in the world, especially in the USA by far, so understanding the transmission pattern of this pandemic has become more urgent. Our analyses of the COVID-19 surveillance data depict remarkably heterogeneous transmission across states during the COVID-19 outbreak in the USA from 15 March 2020 to 15 September 2020. The degree of heterogeneity is characterized by random effects parameter estimates. With the Kendall and Spearman-rank correlation tests, we explore the association between weather/climate factors and daily confirmed COVID-19 cases for each state, which is further used for the analysis of the spatial and temporal occurrence of COVID-19.

Some interesting findings are noteworthy. First, the heterogeneity of COVID-19 transmission across states is observed in all four components, which implies that there are different situations in different states and the same strategies may not work perfectly to contain this pandemic in all states.

Second, some weather/climate factors (i.e., CC, Hu, MaT, MiT, and CDTAO) demonstrate significant correlations with daily confirmed cases in many states. In particular, MaT, MiT, and CDTAO have a strong association with most states. Based on the estimated coefficients from the epidemic-boosted component, one can further find that these variables correspond to daily confirmed cases with a negative correlation in almost all states, i.e., higher MaT, MiT, and CDTAO correlate with less daily confirmed cases. This phenomenon suggests that climate change in the local and adjacent areas could affect the possibility of infection in this area.

Third, since the estimates of the epidemic-within component in most states are predominant, their corresponding values can represent the fluctuation of the daily confirmed cases. Since the estimated coefficients of the epidemic-within component in most states are smaller than one ($\exp\left(\alpha^{\left(\lambda \right)}+V_{r}^{\left(\lambda \right)}\right)<1$), the community-level spread of COVID-19 in most states is remarkably mitigated and the transmission intensity is decreased. Furthermore, we believe that the relatively large number of daily confirmed cases in the current stage are mainly due to the previous large number of infected cases, and the number of new confirmed cases per day will gradually decrease as time goes on.

Fourth, for the future tendency of the daily confirmed cases, the influence of the previous confirmed cases is the most important among the four components. This means that for states like Texas (TX), which has a large number of confirmed cases, the risk of a sharp increase of the daily confirmed cases is still higher than other states with less confirmed cases. Therefore, there is no doubt that regulations like social distancing or wearing masks in public places are clearly necessary.

Final, since the endemic components for some states (e.g., Vermont (VT)) show obvious predominance, other possible variables that could influence COVID-19 transmission need to be further determined.

## Figures and Tables

**Figure 1 ijerph-18-00774-f001:**
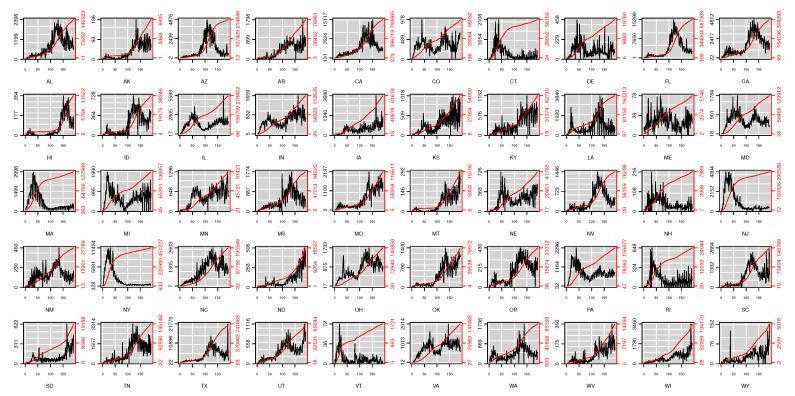
Original data of state-level daily positive test cases in the USA from https://www.kaggle.com/sudalairajkumar/covid19-in-usa (excluding Washington, D.C.). The black line indicates the related daily confirmed cases and the red line the cumulative confirmed cases. The names of all states are denoted by the related postal codes.

**Figure 2 ijerph-18-00774-f002:**
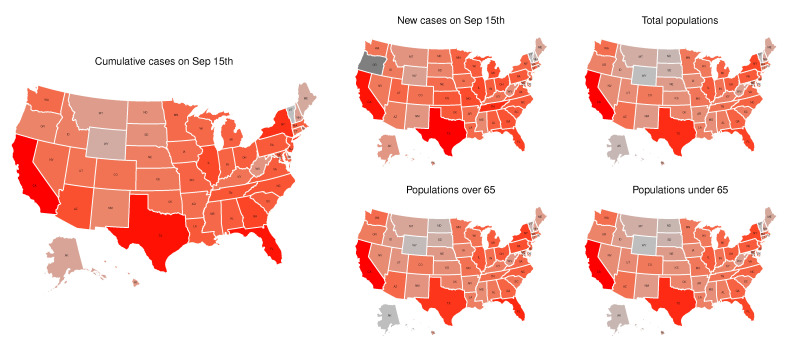
State-level population related information. We took the logarithm of the original data for better visualization.

**Figure 3 ijerph-18-00774-f003:**
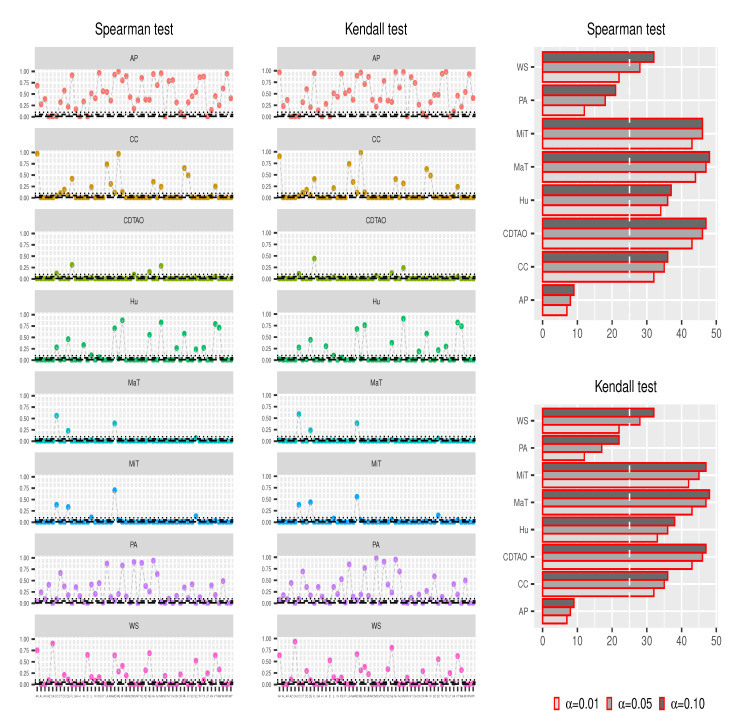
State-level Spearman and Kendall test results. Scatter plots show the related *p*-value, and bar plots show the related number of states that are significantly influenced by related factors with α=0.01,0.05, and 0.10, respectively, which are drawn with dotted black lines. The white dotted lines in the bar plots are equal to 25, which is approximately half of the total number of analyzed states. Maximum temperature (MaT), minimum temperature (MiT), humidity (Hu), the probability of precipitation appearance (PA), the percentage of cloud coverage (CC), sea-level air pressure (AP), wind speed (WS), and the columnar density of total atmospheric ozone (CDTAO)

**Figure 4 ijerph-18-00774-f004:**
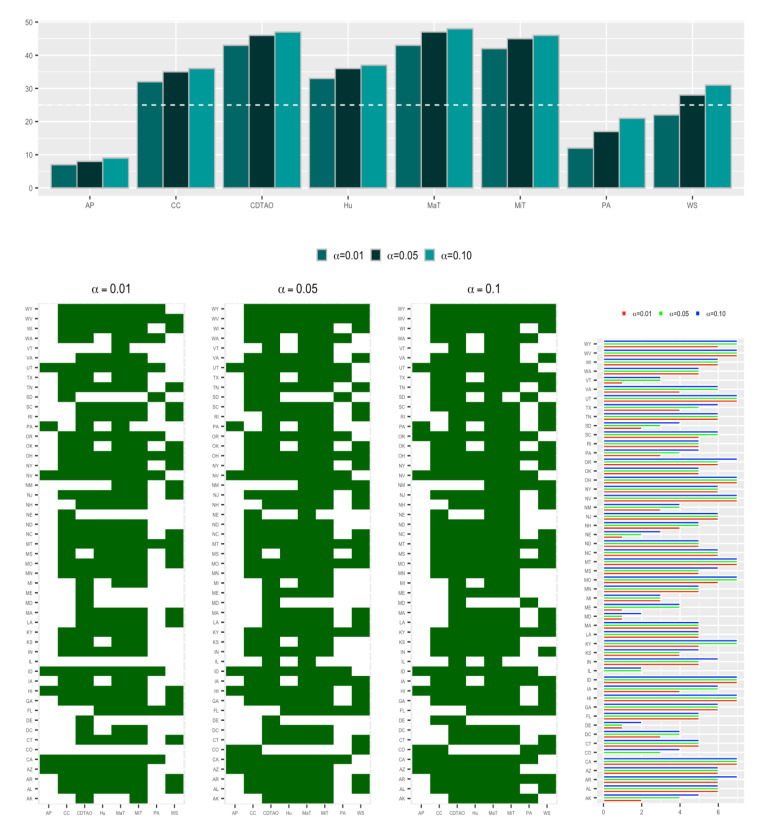
State-level combination of Spearman and Kendall tests. Two-dimensional barcode plots show whether a factor is significantly associated with a state by both the Spearman and Kendall tests (with squares in dark green color). The bar plot represents the number of states that are significantly associated with each factor with α=0.01,0.05, and 0.10, respectively, by both the Spearman and Kendall tests. The horizontal line in the top subplot is equal to 25, which is approximately half of the total number of analyzed states.

**Figure 5 ijerph-18-00774-f005:**
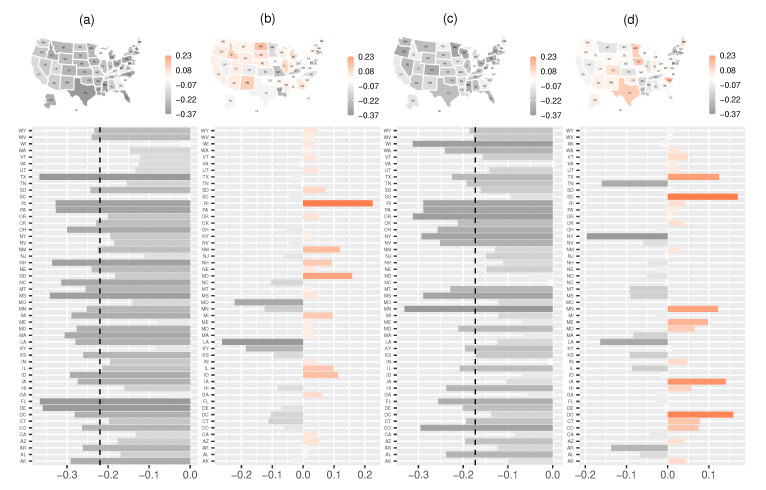
State-level estimated random effects in the multivariate time series model for: (**a**) epidemic-within component $V^{\left(\lambda \right)}$; (**b**) epidemic-between component $V^{\left(\psi \right)}$; (**c**) epidemic-boosted component $V^{\left(\theta \right)}$; and (**d**) endemic component $V^{\left(\zeta \right)}$. Vertical dotted lines in (**a**,**b**) are $-\alpha^{\left(\lambda \right)}$ and $-\alpha^{\left(\theta \right)}$. There is a strong variation in all four components, and different random effects for different states have various influences.

**Figure 6 ijerph-18-00774-f006:**
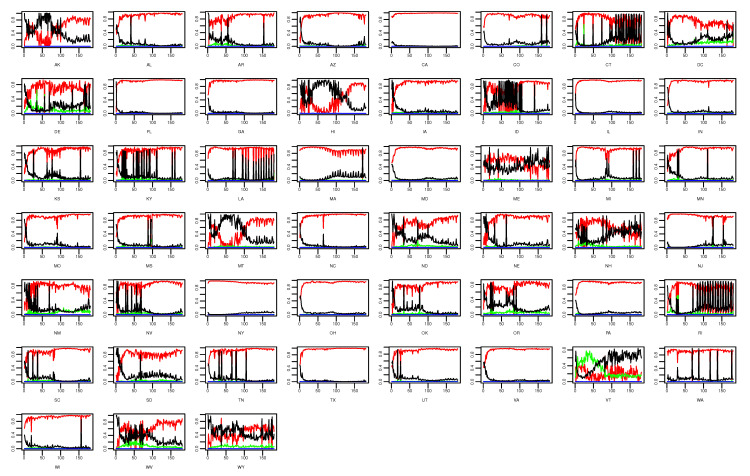
State-level daily percentage of daily confirmed cases with respect to the epidemic-within, endemic-between, epidemic-boosted, and endemic components, which are indicated by red, green, blue, and black lines, respectively.

**Figure 7 ijerph-18-00774-f007:**
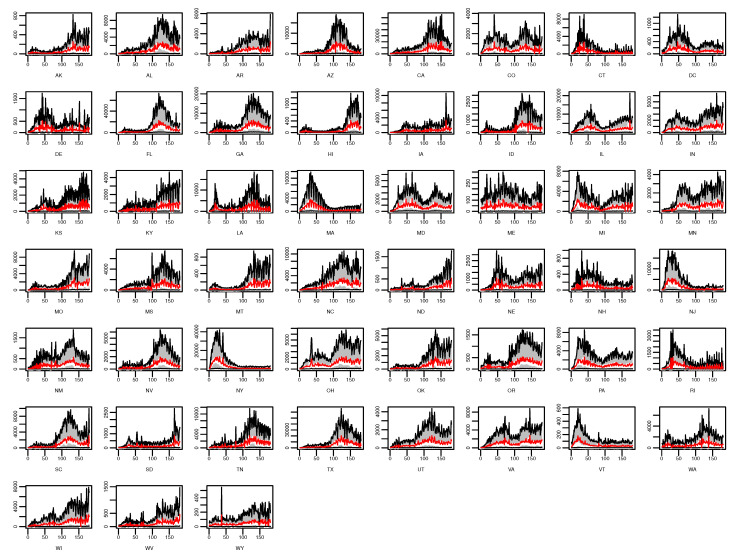
State-level estimated means of daily confirmed cases and related 90% confidence intervals. The gray band represents the 90% confidence interval, the white line the observations of the daily confirmed cases, and the red line the estimated mean values.

**Table 1 ijerph-18-00774-t001:** Different time lags and the corresponding penalized log-likelihood values shown with the mean and standard deviation (sd). Thirty different initial processes for each lag are randomly implemented. The corresponding mean and sd are calculated based on 30 repetitions.

Lags	l=1	l=2	l=3	l=4	l=5	l=6	l=7
$\mathrm{mean}\left(l(\pi ,v)\right)$	47512882.850	48510611.518	48188861.487	47895162.927	47783167.600	47998732.406	47976026.240
$\mathrm{sd}\left(l(\pi ,v)\right)$	1600987.450	301043.787	923652.391	849380.316	1297418.980	825285.208	1055666.220
lags	l=8	l=9	l=10	l=11	l=12	l=13	l=14
$\mathrm{mean}\left(l(\pi ,v)\right)$	47747910.670	47946826.823	47493940.570	47630036.889	47306645.479	47309743.565	47115278.420
$\mathrm{sd}\left(l(\pi ,v)\right)$	1199672.290	643524.434	1288241.920	773951.669	920626.589	673304.655	1158747.060

**Table 2 ijerph-18-00774-t002:** Optimal parameter estimates with time lag l=2. One-hundred different initial processes are randomly implemented. The related mean and standard deviation (sd) are calculated based on 100× outcomes.

Estimates	$\sigma_{\left(\lambda \right)}$	$\sigma_{\left(\psi \right)}$	$\sigma_{\left(\theta \right)}$	$\sigma_{\left(\zeta \right)}$	$\alpha^{\left(\lambda \right)}$	$\alpha^{\left(\psi \right)}$	$\alpha^{\left(\theta \right)}$	$\alpha^{\left(\zeta \right)}$
mean	0.851	0.836	0.816	0.853	0.220	−0.136	0.173	−0.233
sd	0.121	0.113	0.107	0.124	0.911	0.929	0.958	0.991
estimates	$\beta_{1}^{\left(\psi \right)}$	$\beta_{1}^{\left(\zeta \right)}$	$\beta_{2}^{\left(\zeta \right)}$	$\beta_{3}^{\left(\zeta \right)}$	$\beta_{4}^{\left(\zeta \right)}$	$\varepsilon_{AL}$	$\varepsilon_{AK}$	$\varepsilon_{AZ}$
mean	−0.300	0.232	−0.064	−0.119	−0.076	0.731	0.774	0.764
sd	0.873	1.068	1.004	1.180	1.085	0.285	0.286	0.269
estimates	$\varepsilon_{AR}$	$\varepsilon_{CA}$	$\varepsilon_{CO}$	$\varepsilon_{CT}$	$\varepsilon_{DE}$	$\varepsilon_{FL}$	$\varepsilon_{GA}$	$\varepsilon_{HI}$
mean	0.739	0.697	0.754	0.776	0.744	0.732	0.730	0.693
sd	0.286	0.296	0.284	0.270	0.285	0.301	0.275	0.297
estimates	$\varepsilon_{ID}$	$\varepsilon_{IL}$	$\varepsilon_{IN}$	$\varepsilon_{IA}$	$\varepsilon_{KS}$	$\varepsilon_{KY}$	$\varepsilon_{LA}$	$\varepsilon_{ME}$
mean	0.707	0.673	0.781	0.715	0.696	0.751	0.816	0.784
sd	0.253	0.278	0.284	0.285	0.292	0.281	0.296	0.290
estimates	$\varepsilon_{MD}$	$\varepsilon_{MA}$	$\varepsilon_{MI}$	$\varepsilon_{MN}$	$\varepsilon_{MS}$	$\varepsilon_{NJ}$	$\varepsilon_{NM}$	$\varepsilon_{NY}$
mean	0.780	0.658	0.740	0.724	0.758	0.708	0.858	0.780
sd	0.301	0.291	0.296	0.289	0.308	0.291	0.273	0.306
estimates	$\varepsilon_{MO}$	$\varepsilon_{MT}$	$\varepsilon_{NE}$	$\varepsilon_{NV}$	$\varepsilon_{NH}$	$\varepsilon_{NC}$	$\varepsilon_{ND}$	$\varepsilon_{OH}$
mean	0.716	0.757	0.732	0.780	0.776	0.793	0.769	0.710
sd	0.299	0.281	0.298	0.303	0.284	0.273	0.284	0.301
estimates	$\varepsilon_{OK}$	$\varepsilon_{OR}$	$\varepsilon_{PA}$	$\varepsilon_{RI}$	$\varepsilon_{TX}$	$\varepsilon_{SC}$	$\varepsilon_{SD}$	$\varepsilon_{TN}$
mean	0.667	0.732	0.742	0.763	0.812	0.765	0.746	0.729
sd	0.294	0.277	0.306	0.283	0.296	0.279	0.273	0.290
estimates	$\varepsilon_{UT}$	$\varepsilon_{VT}$	$\varepsilon_{VA}$	$\varepsilon_{WA}$	$\varepsilon_{WV}$	$\varepsilon_{WI}$	$\varepsilon_{WY}$	$\varepsilon_{DC}$
mean	0.736	0.742	0.687	0.757	0.805	0.779	0.727	0.791
sd	0.291	0.287	0.327	0.283	0.287	0.300	0.289	0.296

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
