# Peer review of "Analysis of the Spread of COVID-19 in the USA with a Spatio-Temporal Multivariate Time Series Model"

_ijerph, 2021, doi:10.3390/ijerph18020774_

Round 1

Reviewer 1 Report

I would like to congratulate the authors for their interest in researching in this field.

The authors have made an analysis of the Spread of COVID-19 with Spatio-Temporal Multivariate Time Series Model using area, weather or population data.

While the study is interesting, there are some points that need to be corrected.

  1. There are several previous studies of interest that have not been considered. As an example, the investigations of Papastefanopoulos (2020) and James (2020) or Suganya (2020) provide interesting information that can be contrasted with this study to enrich it. I propose to the authors to complete their research with references to these or other previous studies of interest.
  2. The structure presented does not correspond to the model of article proposed in the journal. Anyway, I understand the particular circumstances of this type of research. That is why, I propose to the authors to make next changes.
    1. It is necessary that the sections that develop the model appear in the same chapter. In a conventional article, this would correspond to “Materials and Methods” chapter. I would propose the title "Development of the model" for this chapter although the authors can propose another of their choice.
    2. Authors should separate “Discussion” and “Conclusions” in two different chapters. It would be convenient that, in the Discussion section, authors compare the model proposed with the results of other similar researches such as those exposed in chapter 1 by the authors or those I have proposed previously.
  3. Discussion is not complete. This section should be expanded considerably and then the main results obtained should be included into Conclusions Chapter.
  4. I believe that, throughout the work, they should expose the limitations of the present research, as well as the future lines of action for its development.
  5. Some figures are too small to be adequately interpreted. Specifically, figures 1, 6 and 7 as well as some parts of figures 3 and 4 are too small to be interpreted in print format. I suggest to the authors to enlarge the size of these figures or to divide them in a bigger number of figures.

I take this opportunity to convey to the authors my support in the implementation of the suggestions sent and wish them a happy new year.

Reviewer 2 Report

The manuscript entitled “Analysis of the Spread of COVID-19 in the USA with Spatio-Temporal Multivariate Time Series Model” has characterized the state level spatiotemporal dynamics of on going COVID-19 pandemics in USA and has discussed state-level correlation between COVID-19 risks with various weather/climate factors.  The research has important contribution on understanding COVID dynamics. The methodology applied here is appropriate and results are consistent with previous studies from other countries. The write up is comprehensive did not found significant error in English, though being not native English speaker other reviewer can review the language part more rigorously.  I suggest accepting the manuscript with minor correction. Some suggestion are given below.

  1. The authors have reviewed plenty of literatures related with methods, however, potential weather and environmental factors associated with COVID-19 transmission has not been reviewed substantially in the introduction. As the primary objectives of the manuscript is to highlight the association, I suggest to have more review on this dimension.
  2. Secondly, model description is too long and it is better to annex it as a supplementary file.
  3. Thirdly, the manuscript should be divided into introduction, method, results and discussion explicitly. As there is no clear demarcation between methods and results, it is bit confusing in reading.

Reviewer 3 Report

The topic of the COVID-19 pandemic’s impact in the USA is an important topic and it is very interesting. I commend the authors for their work. However, there are some issues with the manuscript in the current form. The manuscript requires some changes as indicated below. 

  1. Line 15: perhaps remove “with the COVID-19 spread tendency” and replace with “with the tendency of COVID-19 spreading”. 
  2. Line 15: “statistically significant” instead of “significant”?
  3. Line 29: awkward, should be recession instead of “regression” since regression involves statistical modelling.
  4. Line 39: should replace “states” with “states and territories”, see my next comment
  5. Line 40: Puerto Rico is not a U.S. state, it is a U.S. territory.  As written it seems like it is included with U.S. states.
  6. Line 40: watch spelling “larcitegest”
  7. Line 136: awkward, perhaps replace “temperature average” with “average seasonal temperature”?
  8. Line 169: grammatically awkward, perhaps replace “there is a younger trend in COVID-19 infection severity” with “there is a trend of younger people having severe COVID-19 infections”?
  9. Figure 5: it is very small, I recommend increasing the size
  10. Figure 6: it is very small, I recommend increasing the size.
  11. Line 231: awkward remove “Besides” and capitalize “With”
  12. Line 242-243: this is good but more details would be better, e.g. higher humidity, higher average daily temperatures = higher rates of infections?
  13. Line 245: space required between cases.Since

Please revise the manuscript with tracked changes and upload this version so that the reviewer may see the changes clearly.

Round 2

Reviewer 1 Report

Dear Authors.
You have diligently attended to the requirements of the initial review of the article.
We deeply appreciate your efforts.
Best regards.

Reviewer 3 Report

The COVID-19 pandemic’s impact in the USA is an important topic, and it is very interesting. The authors have revised the manuscript per the reviewer’s suggestions.  I recommend the manuscript should be accepted for publication with some minor revisions. Below are some final recommendations:

  1. Line 263: grammatical error. Please remove “Recent researches [18,19] have shown” with “Recent research [18,19] has shown” 
  1. Line 269: there are commas missing.  “cost is therefore high” should be “cost is, therefore, high.”

Thank you for the opportunity to review this work.